# Effect of Combined Forming and Aging Processes on the Mechanical Properties of the Precipitation-Hardenable High-Strength Aluminum Alloys AA6082 and AA7075

**Steffen Lotz [1,2,\*], Emad Scharifi [1]** **, Ursula Weidig [1] and Kurt Steinhoff [1]**

[1] Metal Forming Technology, University of Kassel, Kurt-Wolters-Straße 3, 34125 Kassel, Germany; emad.scharifi@uni-kassel.de (E.S.); ursula.weidig@uni-kassel.de (U.W.); steinhof@uni-kassel.de (K.S.)

[2] Department of Research and Development, METAKUS Automotive GmbH, Fehrenberger Straße 1a, 34225 Baunatal, Germany

\* Correspondence: steffen.lotz@uni-kassel.de; Tel.: +49-(0)561/804-1944

**Abstract:** The recently increasing demand for hot stamped aluminum components in the automotive and aerospace industries explains the necessity of designing efficient and resource-conserving thermo-mechanical processes. Within the thermo-mechanical process, the simultaneous effect of deformation and temperature accelerate the precipitation kinetics. Therefore, this study focuses on the combined effect of forming and aging processes on the mechanical properties of high-strength aluminum alloys AA6082 and AA7075. For this aim, two different thermo-mechanical aging process strategies after solution heat treatment and quenching in a water-dilutable polymer quenchant are proposed. The superpositioning of the forming step is either performed at the beginning or continuously during the aging treatment. The resulting mechanical properties are characterized using tensile tests. With increasing the plastic elongation, there is an increase in yield and tensile strength, which is accompanied by a significant decrease in strain after failure. Both thermo-mechanical aging strategies reveal mechanical properties similar to the conventional T6 peak aged condition with a significant reduction in process time from 24 h to 5 h.

**Keywords:** thermo-mechanical aging; high strength aluminum alloys; precipitation-hardening

## 1. Introduction

The increasing interest in precipitation-hardenable aluminum alloys for automotive and aerospace applications, due to their excellent strength-to-weight ratio, fosters the competition of production methods [1–3]. Therefore, the demand for efficient processing, in order to achieve short cycle times and low manufacturing costs, becomes more important. With regard to the entire production process of components of precipitation-hardenable aluminum alloys, the aging treatment is the most time and energy-consuming step, but it is inevitable for the setting of the desired mechanical properties of full strength. For this reason, many investigations are carried out on the aging behavior of AA6082 and AA7075 as prominent representatives of this class of aluminum alloys. Investigations on the conventional peak aging of the two alloys AA6082 and AA7075 revealed holding times of 8 h at 165 °C for AA6082 and 20 h at 120 °C for AA7075, respectively [4–8]. Further experimental studies [4,6,9,10] on improving the aging treatment investigate the effect of pre-aging and pre-straining on the precipitation kinetics and mechanical properties of precipitation-hardenable aluminum alloys. Granum et al. [11], for example, focus on the effect of pre-straining in the range from 0.5% to 4.0% on the plastic deformation and strain hardening behavior of 6xxx aluminum series after integrated artificial aging at 185 °C for 8 h. According to this study, the ductility increases with the increasing pre-strain without a significant decrease in the yield and ultimate tensile strength. In another study [7] it is reported that the pre-aging of Al-Zn-Mg-Cu at 190 °C for 5 h enhances the material strength,

due to the increased number density of precipitates. Microstructural investigations using transmission electron microscopy (TEM) show the nucleation of very fine η′ precipitates during the initial stage of the pre-aging, which grow in the course of the subsequent main artificial aging treatment. These homogeneously distributed precipitates represent effective barriers to the dislocation motion during plastic deformation leading to the observed high yield and ultimate tensile strength. In contrast, pre-straining creates a decrease in material strength after artificial aging, due to the acceleration of the precipitation kinetics caused by the generation of a high dislocation density within the grains [7,12,13]. Kolar et al. [6], reported for simultaneous aging and forming of AA6060 that significantly decreased aging times are capable to reach peak-aging conditions with even better mechanical properties. The observed, preferred appearance of precipitates in the vicinity of dislocations is a result of the lattice mismatch and the corresponding elastic misfit stress field between dislocation and matrix, which promotes and accelerates the nucleation of clusters and precipitates. Conversely, these accelerations result in shorter aging times.

Taking the observations of the above-presented studies into account, it seems that the acceleration of the aging treatment of precipitation-hardenable aluminum alloys by a combination with a deformation step is a promising solution. Therefore, this study focuses on the investigation of the influence of amount and point in time of the superpositioned deformation on the precipitation kinetics, the precipitation and strain hardening of AA6082 and AA7075. The understanding of these effects is not only of scientific interest but also of industrial importance with respect to the design of efficient materials processing strategies for those aluminum alloys.

## 2. Materials and Methods

### 2.1. Materials

The experimental investigations in this work were carried out on two precipitation-hardenable aluminum alloys: AA6082 (Al-Mg-Si), which is commonly used in automotive applications, and AA7075 (Al-Zn-Mg-Cu), which is used in aerospace industry. The chemical compositions were characterized by optical emission spectroscopy and are listed in Table 1.

**Table 1.** Chemical composition of AA6082 and AA7075.

| Chemical Elements (wt.%) | Si | Fe | Cu | Mn | Mg | Cr | Zn | Ti | Others |
|---|---|---|---|---|---|---|---|---|---|
| AA6082—as-received (AR) | 0.09 | 0.42 | 0.10 | 0.44 | 0.80 | 0.02 | 0.19 | 0.04 | Balance |
| AA7075—as-received (AR) | 0.08 | 0.12 | 1.6 | 0.04 | 2.7 | 0.19 | 5.9 | 0.05 | Balance |

Both alloys were delivered by Austrian Metal AG in the as-received T6 condition as sheets with a thickness of 1.5 mm. Tensile test samples of both alloys were cut by electrical discharge machining with a gauge length of 164 mm for the thermo-mechanical aging experiments. For further examination of the mechanical properties, smaller tensile samples were cut in the rolling direction (RD) and transverse direction (TD) with a gauge length of 9 mm.

### 2.2. Experimental Set-Up and Program

The experimental setup in the present study consists of a roller hearth furnace (Bättenhausen Rollmod) to heat the samples of the precipitation-hardenable aluminum alloys to the corresponding solution heat treatment temperature [8]. Cooling is performed in a water-dilutable polymer quenchant (W + AQ) that allows a high cooling rate with lower thermal distortion compared to water quench. The thermo-mechanical aging is performed by a tensile testing machine (Hegewald and Peschke inspekt 100) equipped with a heating system. Thermocouples attached to the sample surface ensure a constant temperature level during heating.

The hot deformation samples were first solution heat treated (540 °C for AA6082 and 480 °C for AA7075) for 20 min and then quenched in the water-dilutable polymer quenchant. After heating to their specific aging temperature (165 °C for AA6082 and 120 °C for AA7075 [4,8,14]) the thermo-mechanical aging experiments started. As mentioned before, two different strategies were carried out with parameters according to Figure 1. The first strategy consists of straining the samples with different plastic elongations at the very beginning of the artificial aging, named "Forming at the Beginning of Aging (FBA)" while the second strategy is characterized by a continuous simultaneously performed plastic straining during the complete aging named "Simultaneous Forming and Aging (SFA)". The FBA experiments were performed with a cross-head speed of 1 mm/min whilst it was adapted to the respective elongation and aging duration for the SFA experiments.

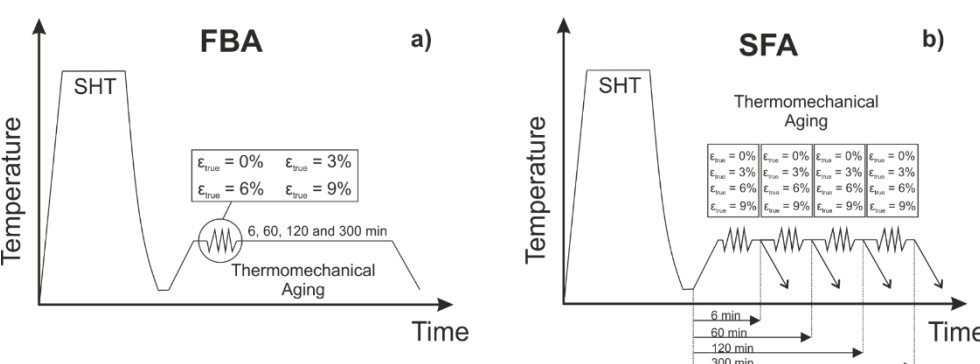

**Figure 1.** Experimental process routes of (**a**) FBA and (**b**) SFA after solution heat treatment and water quench.

## 3. Results

### 3.1. Thermo-Mechanical Aging of AA6082

The influences of thermo-mechanical aging on the final mechanical properties of AA6082 are presented in Figures 2 and 3. Additionally, the ultimate tensile strength (UTS) and yield strength (YS), as well as elongation after fracture for the as-received T6 condition, are indicated. Considering all results, there is no remarkable difference observed between the mechanical properties of the samples in the rolling direction or in the transverse direction. Both treatment strategies, FBA (Figure 2) and SFA (Figure 3), lead at an aging duration of 120 min to strength properties above those of the as-received condition but are accompanied by an elongation after fracture below the as-received condition. The yield strength and the tensile strength increase with increasing plastic strain up to 9% and time up to 120 min. After 120 min of aging, the deformation degree of 9% leads to UTS of 375 MPa and YS of 338 MPa in comparison to 345 MPa and 280 MPa, respectively, for the as-received condition. In contrast, the elongation after failure decreases from 15% of the as-received condition to 10.5% of the highest strength condition, i.e., a strain of 9% and an aging duration of 120 min.

The maximum strength properties are obtained at an aging duration of 120 min. A prolongation of the aging treatment under thermo-mechanical conditions does not contribute to any further significant increase in yield and tensile strength, but tends to a decrease in values for the elongation after failure, Figures 2 and 3.

Concerning the aging strategy of SFA, a premature failure occurred at an aging duration of 300 min with the samples of the targeted strain of 6% and 9% resulting in missing values, Figure 3a–d.

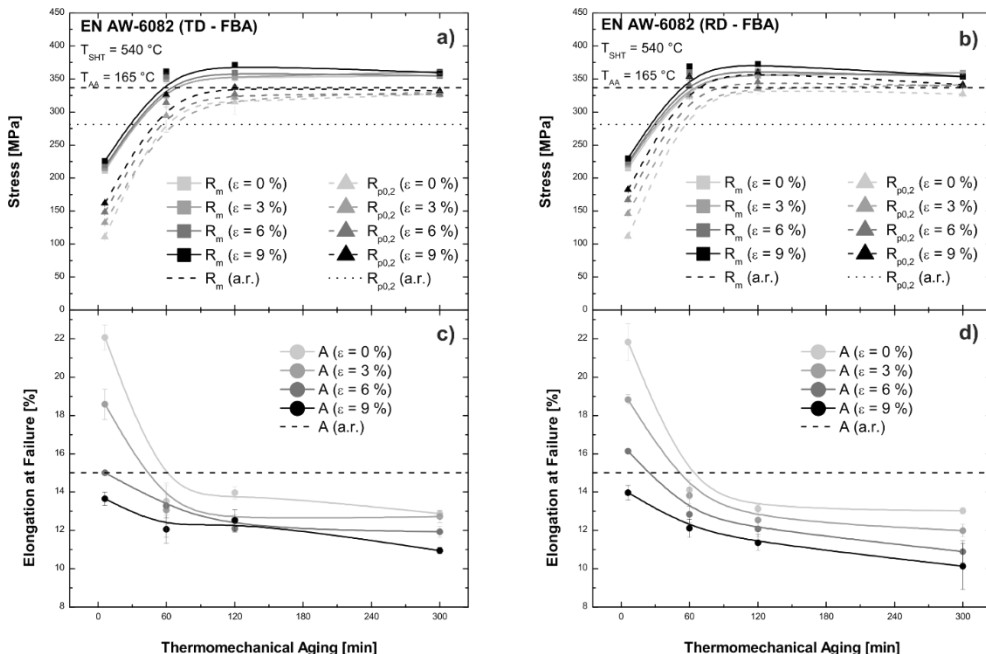

**Figure 2.** Influence of aging duration and strain on yield and ultimate tensile strength of AA6082 for thermo-mechanical aging strategy "Forming before Aging" (**a**,**c**) transverse and (**b**,**d**) parallel to the rolling direction. All specimens are taken from the thermo-mechanically aged conditions and are tested at room temperature.

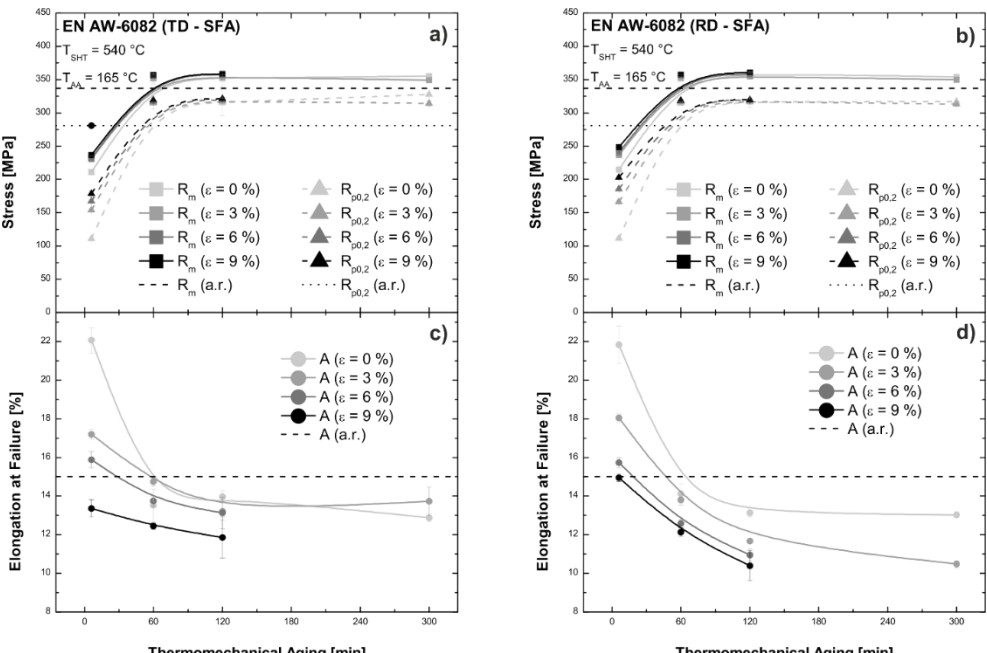

**Figure 3.** Influence of aging duration and strain on mechanical properties of AA6082 for thermo-mechanical aging strategy "Simultaneous Forming and Aging" (**a**,**c**) transverse and (**b**,**d**) parallel to the rolling direction. All specimens are taken from the thermo-mechanically aged conditions and are tested at room temperature.

### 3.2. Thermo-Mechanical Aging of AA7075

Figures 4 and 5 show the results of the experiments with the two thermo-mechanical aging strategies applied on AA7075. As with AA6082, no remarkable difference is observed between the mechanical properties of the samples in the rolling direction or in the transverse direction. The yield and ultimate tensile strength increase with increasing strain up to 9% and aging time up to 300 min, accompanied with decreasing elongation after failure. However, in contrast to AA6082 and regardless the treatment strategy of FBA or SFA, nearly all thermo-mechanical aging experiments result in values of yield and ultimate tensile strength below those of the as-received T6 condition of AA7075. Only an aging duration of 300 min at a strain of 9% leads for both strategies to values of yield and ultimate tensile strength close to the 525 MPa and 585 MPa of the as-received T6 condition. The values of elongation after failure after the thermo-mechanical aging experiments are mainly higher than for the as-received condition with the exceptions of the aging duration of 300 min at a total deformation degree of 9% for the strategy FBA and the aging durations longer than 60 min at deformation degrees of 6% and higher for the strategy SFA. Concerning the aging strategy of SFA, a premature failure occurred at an aging duration of 300 min with the samples of the targeted strain of 9% resulting in missing values, Figure 5a–d.

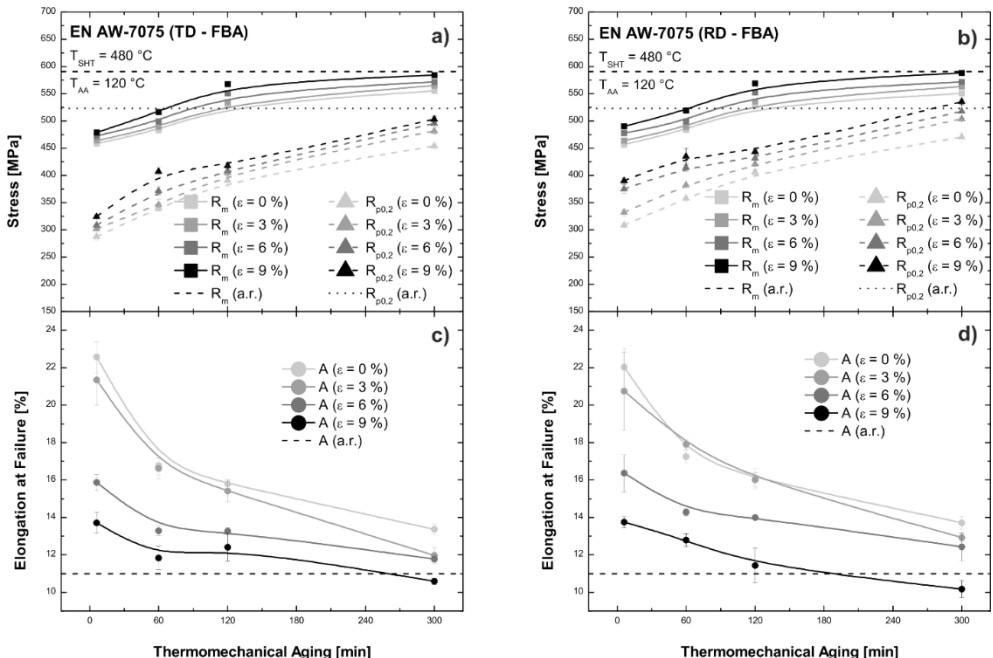

**Figure 4.** Influence of aging duration and strain on mechanical properties of AA7075 for thermo-mechanical aging strategy "Forming before Aging" (**a**,**c**) transverse and (**b**,**d**) parallel to the rolling direction. All specimens are taken from the thermo-mechanically aged conditions and are tested at room temperature.

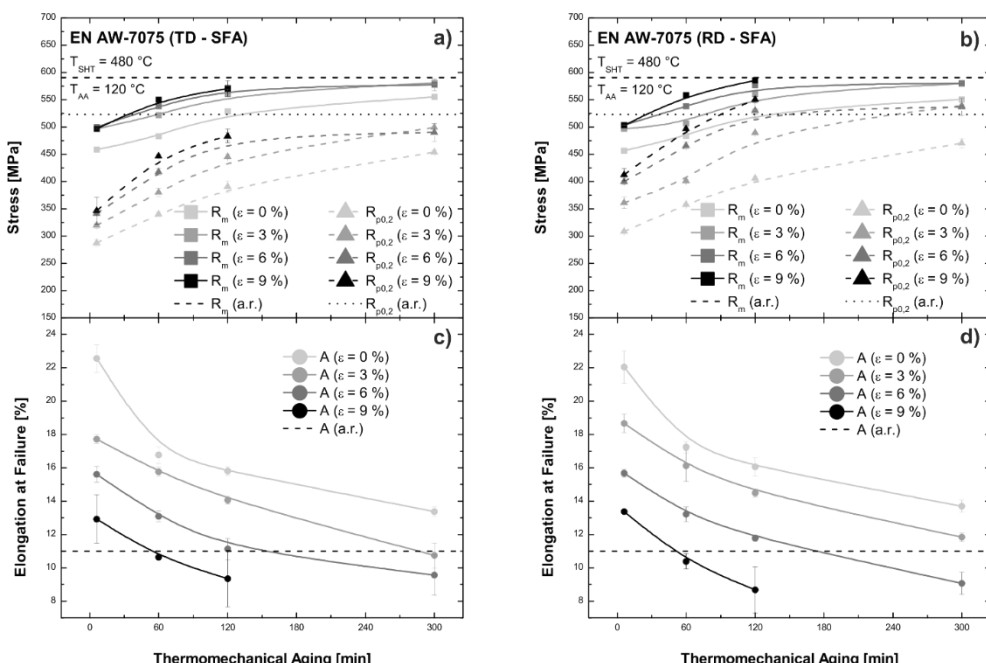

**Figure 5.** Influence of aging duration and strain on mechanical properties of AA7075 for thermomechanical aging strategy "Simultaneous Forming and Aging" (**a,c**) transverse and (**b,d**) parallel to the rolling direction. All specimens are taken from the thermo-mechanically aged conditions and are tested at room temperature.

## 4. Discussion

In conventional precipitation-hardening of aluminum alloys AA6082 and AA7075, the evolution of material strength of precipitation-hardenable aluminum alloys during aging is governed by microstructural mechanisms such as nucleation and growth of precipitates. The number of precipitates is dependent of the precipitate size responsible for the increase in global yield and tensile strength by acting as the main obstacle for moving dislocations. Whilst fine dispersed particles usually result in higher mechanical properties, coarse particles lose their strengthening capacity. Due to the thermal activation of nucleation and growth of precipitates, an optimum aging temperature and time exist, which results in an amount and size of precipitates leading to maximum strength properties [15,16]. Therefore, artificial aging with parameters below but also beyond this optimum leads to decreasing mechanical properties. Increasing the number of dislocations by deformation promotes and accelerates the nucleation that leads to an increasing number of precipitates. At the same aging temperature, the superposition of deformation, therefore, is supposed to shift the peak aging time to a shorter duration. The simultaneous reduction in the exposition to thermal energy is also expected to reduce precipitates growth leading to the finest dispersed particles. This offers the opportunity to reduce aging times at comparable mechanical properties and to gain process efficiency by a thermo-mechanical process approach.

The thermo-mechanical aging experiments in this study on the two precipitation-hardenable aluminum alloys AA6082 and AA7075 generated the following observations and results on the combined influence of forming and aging.

Depending on the alloy, both aging strategies, FBA and SFA, led to mechanical properties comparable to or higher than the T6 condition obtained by conventional treatment. For the alloy AA6082, higher values of yield and tensile strength than for the conventional T6 condition are obtained at an aging temperature of 165 °C. By the superposition of deformation of 9%, a reduction from conventional 8 h to 2 h aging time is possible for both strategies at even superior mechanical strength but correspondingly reduced elongation after failure. Prolonged aging beyond 2 h does not contribute to a further rise in strength properties, due to the thermally activated growth of precipitates. Within the

investigated range of aging time and deformation at an aging temperature of 120 °C, the resulting values of yield and tensile strength are lower for both strategies compared to the T6 condition for the alloy AA7075. With increasing time up to 5 h, the values approach the level of conventional peak aging after 20 h at 120 °C. Taking the long aging duration for conventional processing into account, it may be that despite the accelerating effect of deformation an aging time of 5 h still constitutes conditions of under aging. However, the thermo-mechanical approach proves its capacity for shortening process times for the artificial aging of AA7075. The comparison of the thermo-mechanical aging strategies FBA and SFA reveals no significant differences in the resulting mechanical properties. The acceleration effect of deformation on the nucleation of precipitates is performed either by the increase in dislocation density statically at the beginning or dynamically during the aging treatment.

**5. Conclusions**

Summing up all considerations and results leads to the conclusion that a superposition of deformation to artificial aging is capable to reduce significantly artificial aging times for precipitation-hardenable alloys AA6082 and AA7075. With aging parameters adapted to the accelerated precipitation, due to the increased dislocation density, comparable and even superior strength properties to conventional processing are possible. The thermo-mechanical aging approach opens up new possibilities to design efficient production processes and integration of the investigated process routes into existing forming processes seems sensible. Other investigations in the future should focus on a variety of different aging temperatures to control the growth of precipitates, in order to identify further optimization possibilities. In addition, detailed investigations on the feasibility of common metal forming processes are required for industrial implementation of this previously not extensively investigated process.

**Author Contributions:** Conceptualization, S.L. and U.W.; methodology, S.L. and E.S.; investigation, S.L. and E.S.; writing—original draft preparation, S.L. and E.S.; writing—review and editing, S.L., U.W. and K.S.; visualization, S.L. and E.S.; supervision, K.S. All authors have read and agreed to the published version of the manuscript.

**Funding:** The authors would like to thank the financial support from the Hessen State Ministry for Higher Education, Research and the Arts—Initiative for the Development of Scientific and Economic Excellence (LOEWE) for the project ALLEGRO (Subprojects A2).

**Institutional Review Board Statement:** Not applicable.

**Informed Consent Statement:** Not applicable.

**Data Availability Statement:** Not applicable.

**Conflicts of Interest:** The authors declare no conflict of interest.

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
