# Peer review of "Effect of Combined Forming and Aging Processes on the Mechanical Properties of the Precipitation-Hardenable High-Strength Aluminum Alloys AA6082 and AA7075"

_metals, doi:10.3390/met12081250_

Round 1

Reviewer 1 Report

The present investigation focuses on the acceleration of the aging treatment of precipitation hardenable aluminum alloys by a combination with a deformation step.

This paper represents a nice technical contribution.  It is clearly written and organized. The grammar and sentence structure are fine.  In my opinion, it meets the publishing standards of “Metals”  

However, where the paper is lacking, is on the experimental part, especially on the absence of microstructural characterization.

It is known that the thermo-mechanical procedures in manufacturing processes are one of basic tools for tailoring mechanical properties of metals. Plastic deformation procedures introduces many defects like dislocations or grain boundaries of various types while the heat treatment modifies them ultimately leading to their annihilation. In these types of Alloys, microstructural investigations using transmission electron microscopy (TEM) is required, in order to reveal the nucleation of very fine precipitates during the stages of aging. By increasing dislocation motion during plastic deformation, β“ and η' phases can be observed. This is different from the single-phase precipitates typically observed in these alloy category.

Also Scanning electron microscopy could help the Authors to detect intermetallic phases, such as β-AlFeSi and α-AlFeMnSi intermetallic phases, which are distributed at cell boundaries, connected sometimes with coarse Mg2Si.

Finally, XRD analysis, could detect phases alteration in each case. 

Author Response

Dear reviewer,

Thank you very much for your detailed review and the constructive comments 
regarding our manuscript.

To briefly address your comments:
Unfortunately, our institute of metal forming technology does not have any electron microscopes at its disposal, which is why only the mechanical properties were examined in the underlying study. However, the presented investigations are at the beginning of a detailed study in cooperation with other institutes that have TEM and SEM available. The submitted paper is intended to publish the first results of the very interesting and current topic. The results should therefore form the basis for the following investigations. As you have already criticized, microstructural investigations are of great interest. We are currently in the process of acquiring research partners who can carry out these studies with us.

Sincerely, 

Steffen Lotz

Reviewer 2 Report

1.    This study focuses on the combined effect of forming and aging processes on the mechanical properties of high-strength aluminum alloys AA6082 and AA7075. For this aim, two different thermo-mechanical aging process strategies after solution heat treatment and quenching in a water-dilutable polymer quenchant are proposed. The superpositioning of the forming step is either performed at the beginning or continuously during the aging treatment. The manuscript is very well structured. The discussion of the results and the conclusion allow relevant knowledge to be drawn.

2. It is recommended to add Journal articles in the last five years.

Author Response

Dear reviewer,

Thank you very much for your detailed review and the constructive comments
regarding our manuscript.

To briefly address your comments:

Thank you for your positive feedback. As you correctly mentioned, we have addedrelated articles from the last five years. 

Sincerely, 

Steffen Lotz

Reviewer 3 Report

1) The first thing that attracts attention when reading this article is a very small number of references to previous works. Aluminum alloys in general, and these groups of alloys in particular, have been well studied by scientists for a long time. Please conduct a more detailed review of the available literature.

2) Also noteworthy is the fact that in this article there are no studies of the structure at all, which is not acceptable.

3) The authors did not pay attention to the study of additional material parameters, such as long-term strength, etc.

4) In general, the article presents interesting results for accelerating the production of finished products, but does not contain a sufficient amount of scientific analysis, such as microstructure analysis, including comparison with those obtained by various modes.

Author Response

Dear reviewer,

Thank you very much for your detailed review and the constructive comments
regarding our manuscript.

To briefly address your comments:

Unfortunately, our institute of metal forming technology does not have any electron microscopes at its disposal, which is why only the mechanical properties were examined in the underlying study. However, the presented investigations are at the beginning of a detailed study in cooperation with other institutes that have TEM and SEM available. The submitted paper is intended to publish the first results of the very interesting and current topic. The results should therefore form the basis for the following investigations. As you have already criticized, microstructural investigations are of great interest. We are currently in the process of acquiring research partners who can carry out these studies with us.

Sincerely, 

Steffen Lotz
Corresponding Author

Round 2

Reviewer 3 Report

Thanks to the authors for the answer! I am very sorry that your organization does not have the necessary equipment and instruments to conduct the required amount of research in this scientific field. In my opinion, this work is very interesting and requires more experimental data for publication in this journal. I would recommend authors to apply to lower ranking journals and would recommend rejecting this article. However, unfortunately, my opinion does not solve anything. Best regards